# Combination of Biochar and *Trichoderma harzianum* Can Improve the Phytoremediation Efficiency of *Brassica juncea* and the Rhizosphere Micro-Ecology in Cadmium and Arsenic Contaminated Soil

**DOI:** 10.3390/plants12162939

**Published:** 2023-08-14

**Authors:** Shaoxiong Yao, Beibei Zhou, Manli Duan, Tao Cao, Zhaoquan Wen, Xiaopeng Chen, Hui Wang, Min Wang, Wen Cheng, Hongyan Zhu, Qiang Yang, Yujin Li

**Affiliations:** 1State Key Laboratory of Eco-Hydraulics in Northwest Arid Region of China, Xi’an University of Technology, Xi’an 710048, China; 580231.ysx@gmail.com (S.Y.); manli0815@163.com (M.D.); 15102711614@163.com (T.C.); blank4634@163.com (Z.W.); man_xp@sina.cn (X.C.); wanghui306@xaut.edu.cn (H.W.); wangmin@xaut.edu.cn (M.W.); wencheng@xaut.edu.cn (W.C.); zhy@xaut.edu.cn (H.Z.); 2PowerChina Northwest Engineering Corporation Limited, Xi’an 710065, China; yangqiang@nwh.cn (Q.Y.); liyujin@nwh.cn (Y.L.)

**Keywords:** biochar, *Trichoderma harzianum*, enzyme activity, rhizosphere soil micro-ecology, regulation of plant remediation

## Abstract

Phytoremediation is an environment-friendly method for toxic elements remediation. The aim of this study was to improve the phytoremediation efficiency of *Brassica juncea* and the rhizosphere soil micro-ecology in cadmium (Cd) and arsenic (As) contaminated soil. A field experiment was conducted with six treatments, including a control treatment (CK), two treatments with two contents of *Trichoderma harzianum* (T1: 4.5 g m^−2^; T2: 9 g m^−2^), one biochar treatment (B: 750 g m^−2^), and two combined treatments of T1B and T2B. The results showed *Trichoderma harzianum* promoted the total chlorophyll and translocation factor of *Brassica juncea*, while biochar promoted plant biomass compared to CK. T2B treatment showed the best results, which significantly increased Cd accumulation by 187.49–308.92%, and As accumulation by 125.74–221.43%. As a result, the soil’s total Cd content was reduced by 19.04% to 49.64% and total As contents by 38.76% to 53.77%. The combined amendment increased the contents of soil available potassium, phosphorus, nitrogen, and organic matter. Meanwhile, both the activity of glutathione and peroxidase enzymes in plants, together with urease and sucrase enzymes in soil, were increased. Firmicutes (dominant bacterial phylum) and Ascomycota (dominant fungal phylum) showed positive and close correlation with soil nutrients and plant potentially toxic elements contents. This study demonstrated that phytoremediation assisted by biochar and *Trichoderma harzianum* is an effective method of soil remediation and provides a new strategy for enhancing plant remediation efficiency.

## 1. Introduction

Industrial activities and agricultural production lead to the accumulation of potentially toxic elements (PTEs) in the soil environment, which is a big danger to the environment and human health [1,2]. Cadmium (Cd) and arsenic (As) can be effectively integrated into food chains through soil–plant systems. These two elements are very poisonous. They can damage the skin and raise the chance of getting cancer, according to IARC (International Agency for Research on Cancer) [3,4]. Therefore, effective treatment of Cd- or As-contaminated soils is urgently needed.

Phytoremediation can reduce the level or toxicity of PTEs in the soil environment. Compared to physical (plowing), chemical (soil cleansing), and electrical (electro-extraction) soil remediation methods [5], phytoremediation is a beneficial remediation technique because it prevents secondary pollution and can be used for massive-scale applications [6,7]. Plants with high biomass and good tolerance to PTEs have been discovered for phytoremediation in recent years [8]. *Brassica juncea* is an energy plant belonging to the *Brassica* genus in the cruciferous family that is highly adaptable and can grow in a variety of harsh environments [9,10]. *Brassica juncea* is a perfect plant source for phytoremediation that has been proven because it has high productivity, a well-established root system, and good resistance to a range of PTEs [11,12,13].

Nonetheless, phytoremediation has a drawback of low effectiveness caused by insufficient mass and PTEs accumulation ability. Therefore, investigators in and abroad have committed to combining amendments on phytoremediation in rehabilitating soil environments contaminated with PTEs [14,15]. Biochar was completely verified to be capable of lowering the biotoxicity of PTEs due to its high porosity and ion exchange capacity [16,17]. By boosting microbial activity in PTEs-polluted soils, biochar can improve the extraction of PTEs from *Brassica napus*, as previous research demonstrated [18]. Biochar can also directly contribute to the availability of phosphorus [19] and alleviate plant potassium deficiency by increasing exchangeable potassium in soil [20], as well as improving soil quality [21,22].

Furthermore, plant growth-promoting fungi (PGPF) have been found to greatly benefit plant growth and their ability to resist invasive plant PTEs by their hosts [23,24]. Numerous studies have explored how PGPF, such as *Aspergillus*, *Penicillium*, and *Trichoderma*, associated with their host, are extremely beneficial to plant growth and stress resistance to invasive PTEs [23,25,26]. *Trichoderma harzianum* (TH), known as one of PGPF, has been shown to have high resistance to Cd and Pb in the experimental area with high PTEs contamination [27], with which endomycorrhizae are formed during mutualistic symbiosis with plant roots [28]. Then, endomycorrhizae can provide a variety of services to the host plant, including nutrients providing for plant growth, as well as enhancing abiotic and biotic stress tolerance [29]. Coincidentally, biochar can create appropriate habitats for beneficial PGPF, promote rapid colonization of crop rhizosphere by fungal inoculants, and mitigate biotic and abiotic plant stress [30]. An efficient and novel technique for the sustainable recovery of polluted soils, recognized by many researchers, is the fusion of biochar with functional fungal strains to improve phytoremediation [31,32].

Based on the studies above, we hypothesize that the effectiveness of phytoremediation for soils polluted with PTEs can be improved by applying biochar and TH together. However, there is a lack of research on how soil properties and functions are restored during phytoremediation with biochar and PGPF on soils contaminated with PTEs, and how these properties change dynamically. Therefore, a field trial was conducted by us to evaluate the hypothesis that cooperative interactions occurred between biochar and TH, which promote the phytoremediation of PTEs by *Brassica juncea* (i.e., plant stabilization and extraction). The primary goals of this research are as follows: (1) to examine the joint impacts of biochar and TH on the development of *Brassica juncea*, soil condition, and rhizosphere microbial communities; (2) to uncover the mechanisms behind the combined impact of biochar and TH on the efficiency of phytoremediation, from the viewpoint of plant–microorganism interactions; (3) to determine the coupling mechanisms between soil environmental factors and soil microbial factors; (4) to determine the crucial factors and mechanisms that influence the enhancement of phytoremediation in mustard plants by biochar and TH.

## 2. Results

### 2.1. Plant Growth and PTEs Content

After 26 weeks’ growth of *Brassica juncea*, the Cd contents both in the shoot and root were decreased with the application of biochar compared to that in the control treatment, while TH enhanced the Cd uptake both in the root and shoot, which is beneficial for Cd transferring from soil to aboveground parts (Figure 1). The highest Cd uptake by *Brassica juncea* was found in the T2B treatment, of which the Cd contents in the root and shoot were increased by 35.30% and 41.17% compared to that in the control (Figure 1A). It could also be found that all treatments showed a greater As uptake compared to that in control (Figure 1B). Combined biochar and TH increased PTEs accumulation and translocation factor in *Brassica juncea* (Figure 1C). The combined treatment produced the greatest biomass of *Brassica juncea*, in which the biomass of shoot and root were increased by 125.85% to 242.18% and 103.13% to 185.94%, respectively. The total amount of chlorophyll increased significantly under the treatment of biochar and TH (Appendix A).

### 2.2. Soil PTEs Content and Availability

Figure 2A,B depict the analysis of samples for T-Cd and T-As from different soil layers. They were taken to assess the impact of six treatments on the PTEs content in the soil. In comparison to the control treatment, the elimination of PTEs in the root zone soil was improved by biochar and TH; furthermore, the combined treatment of biochar and TH showed a better remediation effect. For T2B treatment, the contents of Cd and As decreased by 28.54% and 38.76% in the rhizosphere soil, 27.73% and 40.74% in the 0–15 cm deep soil layer, 49.64% and 53.77% in the 15–30 cm deep soil layer, and 19.04% and 51.30% in the 30–45 cm deep soil layer. For T2B treatment, the available Cd content in the rhizosphere soil was significantly increased by 21.06% compared to the single application of biochar (Figure 2C). The morphological distribution of As in rhizosphere soil and a surface layer of 15 cm soil showed similar results compared to that in control, in which the Ca-As ratio increased significantly and As activity decreased (Figure 2D,E).

### 2.3. Enzyme Activity and Soil Physicochemical Properties

Different soil physicochemical properties after various remediation treatments were observed in Figure 3 and Appendix A. All soil layers had a weakly alkaline pH, with values ranging from 8.04 to 8.89, as shown in Appendix A. Both TH and biochar promoted the release of AN, AP, and AK in the root zone soil and the 15 cm soil layer. Meanwhile, biochar positively promoted OM content in all soil layers. The soil enzyme activities, such as urease, ALP, and sucrase in the rhizosphere, varied greatly. Compared to the control, TH significantly increased the enzyme activities in the rhizosphere soil (Figure 3A–C). Combined treatment increased the activities of urease and sucrase by 87.25% to 97.79% and 101.60% to 111.20%, respectively (Figure 3A,B). Biochar had a promoting effect on the activities of urease and sucrase but reduced ALP activity by 13.75% to 25.62% (Figure 3C).

During the phytoremediation of PTE-contaminated soils, changes in plant enzyme activity were also observed (Figure 3D,E). Both biochar and TH enhanced the POD activity and GSH content in plant shoots and roots compared to that in the control (Figure 3D,E). Concurrently, the root GSH content reached its maximum in the T2B treatment (Figure 3E). Both biochar and TH reduced the MDA content in plant shoots and roots, and the lowest MDA content in shoots and roots was observed in the T2B treatment, with reductions of 75.12% and 7.69%, respectively (Figure 3F).

### 2.4. Diversity and Community Structure of Rhizosphere Soil Microbial Community

Through high-throughput sequencing, all soil samples gave a total number of 341,197 effective bacterial gene sequences and 435,504 effective fungal gene sequences. The bacterial sequences were grouped into 12,816 Operational Taxonomic Units (OTUs), while the fungal sequences were grouped into 4628 OTUs. The sequencing data reflect the bulk of microbial diversity information in the remediation treatment sufficiently as the rarefaction curve nears saturation (Appendix A). According to the Shannon and Simpson indices, the alpha diversity indices, as well as the bacterial and fungal populations, increased in the B and T1 treatments. However, As the application rate of TH increased, soil bacterial and fungal diversity decreased, with bacterial diversity being greater than fungal diversity, as shown in Appendix A.

The taxonomic classification of the entire qualified OTUs has been determined to include 80 different bacterial phyla and 16 fungal phyla. Specific physicochemical parameters and PTEs concentrations from different remediation treatments can affect microbial populations. In all treatments, Proteobacteria, Acidobacteriota, and Firmicutes were the prevalent bacterial phyla, constituting 76.65–87.10% of the total phylum level. The relative proportion of Proteobacteria varied between 21.29% and 57.06% (Figure 4A). The ternary plot analysis showed that Proteobacteria contributed the largest proportion to the ratio in the combined amendment, which exceeded 60% (Figure 4C). For the fungal community, Ascomycota is the dominant fungal phylum, with a relative abundance of 33.85–80.24% (Figure 4B). The proportion of Ascomycota in the combined treatment exceeded 50% (Figure 4D). The Proteobacteria phylum had 11 genera classified as belonging to it. The Firmicutes phylum had eight genera classified as belonging to it. The clustering analysis showed that there was good consistency between the B and T2B treatments in the bacterial community. The dominant bacterial genera in the soil under the T2B treatment are *Bacillus*, *Lechevalieria*, and *Pseudomonas* (Appendix A). The Ascomycota phylum had 22 genera classified as belonging to it. In the treatment of TH and combined treatment, the predominant fungal genera were *Trichoderma*, *Penicillium*, and *Gibberella* (Appendix A).

### 2.5. Responses of Phytoremediation to Microhabitat Properties of Rhizosphere Soil

In order to assess the impact of soil characteristics and PTEs levels on microbial populations in polluted farmland, a redundancy analysis was conducted (Figure 5). According to RDA, the structure of bacterial and fungal communities was mainly influenced by factors such as T-Cd, T-As, CEC, OM, AP, and AK (*p*-value < 0.05). The structure of fungal communities was significantly impacted by A-Cd, with a *p*-value of less than 0.001 (Figure 5A,B). Procrustes analysis was further performed to investigate the correlations between different soil environmental factors and soil microbes (Figure 5C,D). The results demonstrated that the overall variation in environmental factors was considerably uniform among bacterial communities.

In order to investigate the associations between soil properties and microbial communities, a Spearman correlation heatmap combined with UPGMA cluster analysis was utilized (Figure 6). Soil pH did not have a significant effect on the abundance of the bacterial and fungal phyla. However, the soil properties of CEC, OM, AN, AP, and AK had a significant positive association with Firmicutes (a dominant bacterial phylum) and Ascomycota (a dominant fungal phylum). The soil Cd and As exhibited a substantial inverse correlation with Acidobacteriota and Firmicutes (dominant bacterial phyla), as well as Ascomycota (a dominant fungal phylum).

The association between plant characteristics and rhizosphere soil ecological factors was measured by Pearson correlation analysis (Appendix A). The various plant and soil parameters were divided into two groups: Group A (pH, CEC, AN, AP, AK, OM, Shoot Cd, Root Cd, Shoot As, Root As, Shoot POD, Root POD, Shoot GSH, Root GSH, Urease, and Sucrase) and Group B (Shoot MDA, Root MDA, Cd, As, and ALP). Based on the Pearson correlation analysis, the factors of Group A and Group B demonstrated a pronounced negative link with each other, while the environmental factors within each group revealed a considerable positive connection with each other.

The levels of Cd and As in plants were inversely related to PTEs levels in various soil layers. In contrast, the levels of Cd and As in rhizosphere soil had a direct relationship with A-Cd and A-As (Figure 7A). Soil physical and chemical characteristics positively influenced the functions of soil urease and sucrase, whereas soil PTEs adversely affected them, with A-Cd and A-As being the most important factors (Figure 7B). The accumulation of Cd and As in *Brassica juncea* was found to be inversely related to soil PTEs levels, with T-Cd and T-As being the most crucial factors (Figure 7C). The correlation of MDA activity in plants with soil environment was opposite to that of POD and GSH, with T-As, A-As, and A-Cd being the most critical factors (Figure 7D).

## 3. Discussion

### 3.1. Soil PTEs Content and Plant PTEs Accumulation

Phytoremediation was an effective method in promoting the transport of PTEs in soil layers at different depths, transforming availability to organisms of PTEs in the rhizosphere soil [33]. PTEs transformation can be facilitated by biochar into less available forms in soil without reducing the whole quantity of PTEs [34], and the poisonousness of Cd and As to soil microbes and plants was mitigated by biochar [35]. The combination of biochar and TH significantly reduced T-Cd and T-As content in soils at different depths during phytoremediation, with the greatest reduction observed in soil layers at 30 cm and 40 cm depths (Figure 2). Additionally, greatly improved performance of plant-assisted cleanup for PTEs-contaminated soil was achieved through the synthesis of biochar and TH treatment. The formation of overall chlorophyll and the accumulation of biomass in the plants was enhanced by TH (Appendix A) [36,37]. The symbiotic relationship of endophytic fungi had various benefits for plants, such as the elevated production of phytohormones and the maintenance of the internal hormone equilibrium [38]. TH enhanced antioxidant enzyme activities and improved glutathione content in mustard. The plant’s capacity to transport and accumulate PTEs can be strengthened by altering its cellular mechanisms and biological processes [29,39,40]. The concentration of PTEs in plants demonstrated a significant negative link with the PTEs content in the soil. However, a remarkable positive association was discovered between the PTEs content in the inter-root area and the PTEs content in each soil layer (Figure 7A). The above findings mean that the combined amendment method can encourage the movement of cadmium from deeper soil layers to the surface. Additionally, it can significantly increase the PTEs accumulation of *Brassica juncea,* and it promotes the migration of cadmium and arsenic from plant roots to shoots.

### 3.2. Plant Stress Tolerance and Soil Physicochemical Properties

The implementation of biochar had a beneficial impact on AN, AP, and AK in rhizosphere soil and the top 15 cm soil layer, as well as on OM content in each soil layer (Appendix A). It is typically believed that soil pH has a substantial impact on how accessible soil PTEs are [41]. However, since the pH was not improved in this study, the availability of soil PTEs was affected by other factors. OM was shown to be adversely associated with soil Cd and As (Appendix A), which means the increased soil OM content inhibited soil PTEs activity and reduced PTEs toxicity [42,43]. In this study, the combined treatment was found to heighten the performance of urease and sucrase enzymes while simultaneously curbing ALP activity. Based on the previous study, ALP activity has a significant relationship with soil AP. In this study, AP contents were improved greatly, especially in the treatment with biochar. ALP gene expression is induced in phosphorus-poor conditions, and ALP expression is inhibited by increases in AP content [44]. The interaction of PTEs ions with enzyme thiol groups may also play a part in the suppression of soil enzyme activity caused by PTEs [41]. The activity of the enzyme’s urease and sucrase in the soil correlated favorably with soil pH, CEC, OM, AN, AP, and AK, where AN and AP were the most significant factors. However, soil urease and sucrase enzyme activity exhibited a negative correlation with soil PTEs, where A-Cd and A-As were the most significant factors. Enhancing the activity of urease and sucrase enzymes, biochar, and TH increased soil nutrients levels and minimized the effects of PTEs [45]. Reactive oxygen species (ROS) can be greatly increased once PTEs have entered the plant tissues, producing cellular membrane damage and antioxidant enzyme deactivation, resulting in plant tissue damage and even death [46]. Biochar and TH reduced MDA activity in plants and significantly increased POD and GSH activities (Figure 3), which mitigated Cd and As-induced oxidative damage. The increase in POD and GSH activities was more pronounced in the roots than in the shoots, and this is related to TH’s capacity to encourage *Brassica juncea*’s uptake of PTEs [47]. GSH plays a defensive role by chelating PTEs and detoxifying foreign pollutants, which can reduce the damage of total PTEs to plants [48]. The increase in GSH content helps plants to accumulate PTEs, thereby improving the efficiency of remediation. A favorable relationship was found between soil nutrient levels and POD activity (Figure 7D). The improvement of soil nutrient status helps to improve the growth and enzyme activity of plants [49].

### 3.3. Microbial Community Diversity and Structure Differences

The biological diversity and organization of soil organisms can be significantly impacted by PTEs contamination, which can have a dramatic impact on the physical and chemical characteristics of soil [50]. In the fungal community, the composition of microorganisms became more concentrated. TH inhibited the proliferation of other fungi in the soil by colonizing the plant rhizosphere [51]. But with increasing TH application rate, bacterial and fungal diversity within soil declined. The main cause is that TH made soil A-Cd and A-As concentrations higher; as a result, some of the microorganisms that could not tolerate PTEs died. The soil microbial quantity and diversity then decreased [52,53]. Proteobacteria, as the most abundant bacterial phylum in the soil environment, has been reported to be highly tolerant to PTEs in harsh conditions [54,55]. Under the combined treatment, the dominant bacterial genera were *Bacillus*, *Lechevalieria*, and *Pseudomonas*. These species are very competitive and essential for PTEs tolerance and the establishment of plants [56]. The fungal phylum Ascomycota showed adaptability to harsh environments and exhibited a wide distribution in areas with serene PTEs pollution [57,58]. Ascomycota is one of the largest phyla in the fungal kingdom [59] and possesses complex spores, which can mitigate PTEs stress by immobilizing or accumulating PTEs within their bodies [60]. Under the combined amendment, the predominant fungal genera were *Trichoderma*, *Penicillium*, and *Gibberella*. *Trichoderma* and *Penicillium* have strong abilities in accumulating and transporting PTEs, which helps to improve the adaptation to various harsh environments [57].

### 3.4. Modulation Mechanisms between Phytoremediation and Rhizosphere Microecology

Phytoremediation is closely related to the ecological characteristics of the rhizosphere, including soil physicochemical properties, PTEs contents, and microbiome [61]. Soil physicochemical properties (CEC, OM, AN, AP, AK) and T-Cd and T-As concentrations were the primary elements influencing the makeup of bacterial and fungal populations. Additionally, the key drivers of the composition of the soil microorganisms were soil nutrients, including AN and AP [62]. The predominant phylum of soil bacteria, Acidobacteriota, exhibits an extremely unfavorable association with plant Cd concentration (Figure 5). Previous studies have demonstrated a similar result, showing that enrichment of Acidobacteriota can decrease Cd concentration in plants and enhance the uptake of mineral elements by plants [63]. The number of PTEs in plants was positively correlated with the bacterial phylum Firmicutes, and a competitive relationship between Acidobacteriota and Firmicutes in the dominant bacterial species was found [64,65]. In addition, contents of both Cd and As in the rhizosphere were negatively correlated with the dominant bacterial phylum Firmicutes, indicating that Firmicutes are sensitive to the contents of Cd and As [55]. Additionally, it has been demonstrated that Firmicutes have several genes for PTEs susceptibility that help them to tolerate high amounts of PTEs pollution [66]. Cd and As contents of soil revealed both adversely associated with the dominant fungal phylum, Ascomycota. However, Cd contents both in root and shoot have a significantly positive correlation with Ascomycota. This fungal phylum acts as an endophytic fungus, which can reduce PTEs cytotoxicity in roots and enhance PTEs absorption [67]. The combined treatments showed significant changes in the diversity and bacterial and fungal community structure in the soil. Dominant bacteria and fungi can compete for available resources and also cooperate with each other to form stable coexisting communities. The complex relationship mentioned above is the main reason for improving the soil rhizosphere microecology [45] and promoting the enrichment efficiency of *Brassica juncea* for PTEs.

## 4. Materials and Methods

### 4.1. Materials

The experiment was conducted from March 2021 to November 2022 in Xi’an, Shaanxi, China (34°18′43″ N, 108°54′31″ E). A warm temperate monsoon climate prevails in the area. It has an annual rainfall of 609.6 mm, and the average temperature during the study period is 23.3°. The soil texture is silty clay loam (18.22% clay, 45.98% silt, and 35.80% sand). Large areas of soil pollution in the experimental area were caused by prolonged irrigation of sewage. The initial physicochemical characteristics of the contaminated soil, of which Cd and As were chosen as the remediation targets, are summarized in Appendix A.

Seeds of *Brassica juncea* were procured from Gu Yue Flower and Plant Seeds Business Department of Quanzhou County, Shandong Province. All the seeds were disinfected using a 2% sodium hypochlorite solution for 30 min, and any remaining solvents on the seed surface were then washed away with three rinses in disinfected filtered water. The microbial agent TH was obtained from Shandong Lvlong Biological Technology Co., Ltd. (Jinan, China) with an effective bacterial count of ≥1 billion/g. The biochar used in this experiment was purchased from Zhenjiang Runwu Environmental Science and Technology Co., Ltd. (Wenzhou, China). The main properties of biochar are listed in Appendix A.

### 4.2. Experimental Design and Sample Collection

Biochar and TH were chosen as the amendment to improve *Brassica juncea.* The application rate of biochar (B: 750 g m^−2^) and TH (T1: 4.5 g m^−2^ and T2: 9 g m^−2^) were determined based on previously reported studies [68]. Six treatments were established for the field experiments: (1) *Brassica juncea* treatment alone (CK), (2) *Brassica juncea* + biochar treatment (B), (3) *Brassica juncea* + 4.5 g m^−2^ TH treatment (T1), (4) *Brassica juncea* + 9 g m^−2^ TH treatment (T2), (5) *Brassica juncea* + biochar + 4.5 g m^−2^ TH treatment (T1B), and (6) *Brassica juncea* + biochar + 9 g m^−2^ TH treatment (T2B). Six treatments were applied in randomized plots with three replicates. 

Each plot has an aggregate area of 4 m^2^ with dimensions of 2 m in length and 2 m in breadth. To minimize interference, a 50 cm wide isolation strip was placed among the plots. Before implementing the soil pollution treatment, a 30 cm depth of deep plowing was accomplished. After three days, biochar was spread on the designated treatment plots in the plowed layer soil of 0–15 cm. Then, seeds were sown after three days, and the microbial treatment was sprayed according to the experimental setup after *Brassica juncea* planting.

After 26 weeks of the growth period, *Brassica juncea* and soil were sampled. A portion of the plant samples was used to determine plant biomass and PTEs concentrations. Plant samples were oven-heated to 105° for 30 min to inactivate biological activity before drying to consistent weight at 80° for biomass measurements. Another portion of the samples was stored in liquid nitrogen for subsequent enzyme activity assays. Based on the growth characteristics of plant root systems, the rhizosphere soil and 0–15 cm, 15–30 cm, and 30–45 cm of soil were sampled for the study. The communities of soil microbes were examined using rhizosphere soil. The remaining samples were cleaned in the lab, allowed to air dry, and then stored in Zip-lock bags. The air-dried soil samples were sieved through a 1 mm nylon sieve for organic matter, available phosphorus, available potassium, and alkaline nitrogen and then sieved through a 0.15 mm nylon sieve for PTEs.

### 4.3. Soil and Plant Physicochemical Properties and Enzyme Activities

A pH-automated analyzer (Mettler Toledo, Greifensee, Switzerland) was used to measure the pH of the soil at a 1:2.5 (*w*/*v*) soil-to-water ratio [69]. The volumetric potassium dichromate method with external heating was implemented to measure the amount of soil organic matter (OM) [70]. Available phosphorus (AP) was measured using a Smart Chem automatic discontinuous chemical analyzer (SmartChem450, AMS Alliance, Courtaboeuf, Italy) [71]. Atomic absorption spectrophotometry was adopted to quantify the amount of available potassium (AK) [72]. The alkali diffusion method was implemented to measure the soil’s alkali hydrolyzable nitrogen (AN) [73]. Ammonium chloride–acetic acid exchange was implemented to determine the cation exchange capacity (CEC) [74].

Urease was determined utilizing the phenol–sodium hypochlorite colorimetric technique [75], alkaline phosphatase (ALP) was measured with the sodium phosphate colorimetric method [76], and sucrase was quantified utilizing the 3,5-dinitrosalicylic acid colorimetric technique [77]. Enzyme activities both in plant shoots and roots were also determined. Sedlak and Lindsay’s approach was adopted to determine the glutathione (GSH) content [78]. Enzyme solutions were prepared in pre-cooled mortar, and plant malondialdehyde (MDA) and peroxidase (POD) contents were determined using a UV-1800 spectrophotometer (Aoyi Instruments (Shanghai) Co., Ltd., Shanghai, China) [79].

### 4.4. Potentially Toxic Elements Analysis

The four acids, HNO_3_, HCl, HF, and HClO_4_ (5:4:3:2, *v*/*v*), were applied to dissolve the soil samples in digestion tubes on a graphite digestion instrument (EH-2542, Beijing Auwii Instrument Co. Ltd., Beijing, China). The total Cd (T-Cd) [80], available Cd (A-Cd) [81], and total As (T-As) [82] content were determined by atomic absorption spectrophotometry. Atomic absorption spectrometry (GF-AAS) was used to measure the quantity of Cd, whereas F-AAS was used to assess the quantity of As. Meanwhile, according to the extraction procedure described in the literature, the concentrations of A-As, Fe-As, and Ca-As in soil were extracted and determined by AFS [83]. The determination of PTEs content in plants is divided into two parts, including root Cd, shoot Cd, root As, and shoot As for both aboveground and underground parts of the plant. In a graphite digestion apparatus, the plant materials are digested using a 7:1 *v*/*v* solution of HNO_3_ and HClO_4_, and the PTEs content is measured using atomic absorption spectrophotometry [82,84]. Two licensed reference materials were employed for quality control and assurance analyses: Chinese soils (GBW07401, Institute of Geophysical and Geochemical Exploration (IGGE), Beijing, China) and laver (GBW08521, National Institute of Metrology of China (NIM), Beijing, China).

### 4.5. Potentially Toxic Elements Translocation and Accumulation

To assess the efficacy of phytoremediation, we estimated the accumulation of PTEs per plant (in μg plant^−1^) and the translocation factor (TF) [85]. In these equations, PTEs_shoot_ and PTEs_root_ denote the amounts of Cd or As in the shoot and root, respectively, while B_shoot_ and B_root_ indicate the shoot and root mass, respectively.
TF = PTEs_shoot_/PTEs_root_
PTEs accumulation (μg plant^−1^) = PTEs_shoot_ × B_shoot_ + PTEs_root_ × B_root_

### 4.6. DNA Isolation, Sequencing, and Microbial Diversity Analysis

DNA from rhizosphere soil was extracted. PCR was used to amplify and sequence the 16S rRNA and 18S rRNA genes of the bacteria and fungi, respectively. Following the procedure outlined by Gao et al. [86], primers ITS5-1737F (5′-GGAAGTAAAAGTCGTAACAAGG-3′) and ITS2-2043R (5′-GCTGCGTTCTTCATCGATGC-3′) were used to amplify fungal DNA targeting the fungal ITS region. PCR amplification of the V3–V4 variable region of the bacterial 16S rRNA gene was performed using the universal primers 515F (5′-CCTAYGGGRBGCASCAG-3′) and 806R (5′-GGACTACNNGGGTATCTAAT-3′). The TruSeq^®^ DNA PCR-Free Sample Preparation Kit (Illumina, San Diego, CA, USA) was applied for creating sequencing libraries, and index codes were added to the libraries as instructed by the manufacturer. The 18S rRNA and 16S rRNA amplicon high-throughput paired-end sequencing was performed using Illumina (Kapa Biosciences, Woburn, MA, USA) NovaSeq6000 platform. The NCBI Sequence Read Archive (SRA) database has accepted the raw sequencing data from this study and assigned accession numbers for fungi and bacteria (PRJNA936863 and PRJNA936852, respectively).

### 4.7. Statistical Analysis

Data statistics and analysis were performed using IBM SPSS Statistics 26. Duncan’s multiple range test was employed to calculate significance at a level of *p* < 0.05. SigmaPlot 14.0 was used for the quantification and plotting of soil and plant-related indicators. The R programming language (http://www.r-project.org, accessed on 1 May 2023) was used to investigate the probable connection between microbial populations and soil parameters of the environment. The correlation analysis approach of redundancy analysis (RDA) was utilized to determine the important environmental factors that influence microbial community structure.

## 5. Conclusions

Compared to the single amendment, the biochar and *Trichoderma harzianum* combined application considerably improved the phytoremediation efficiency of *Brassica juncea* and enhanced the rhizosphere microbial community and soil quality. The main results were as follows: Biochar significantly contributed to the accumulation of biomass in *Brassica juncea*, while the activity of antioxidant enzymes and PTEs transport was supported by TH. Biochar and TH synergistically enhance the biomass accumulation and PTEs transport of *Brassica juncea*. This results in increased accumulation of Cd and As in plants, thereby improving plant remediation efficiency. Using TH and biochar together decreased the MDA in plants and ALP in the soil while increasing the GSH, POD, soil urease, and sucrase. Additionally, the combined biochar and TH treatment increased AN, AP, and AK in rhizosphere soil and the top 15 cm soil layer, as well as on OM content in each soil layer by biochar. Meanwhile, this decreased the levels of Cd and As in all soil layers. Simultaneous application significantly enriched bacteria (mainly including Proteobacteria, Acidobacteriota, and Firmicutes) and fungi (Ascomycota) that promoted plant resistance to PTEs, and improved plant growth. RDA analysis and Spearman correlation analysis showed that soil physicochemical properties (CEC, OM) and PTEs (T-Cd, T-As) contents were the main factors affecting the structure of bacterial and fungal communities. Thus, the combined application of biochar and TH is highly recommended as an effective approach to improve the phytoremediation efficiency of *Brassica juncea* on Cd- or As-contaminated soil. This study provides new insights into the response mechanism of the combined application of biochar and Endophytic fungi (*Trichoderma harzianum*) in regulating the phytoremediation efficiency and rhizosphere microbial communities of soil. Further research is needed to explore the mechanisms of promoting plant remediation by combining biochar and PGPF.

## Figures and Tables

**Figure 1 plants-12-02939-f001:**
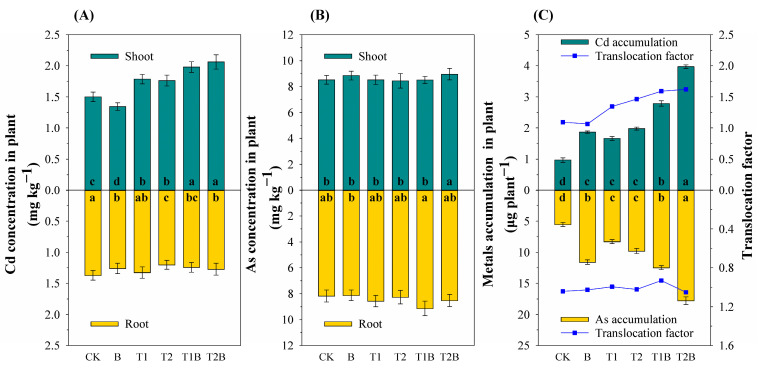
The concentration of Cd (**A**) and As (**B**) in the shoot and root of *Brassica juncea*. (**C**) Plant PTEs accumulation and translocation factor. Different letters represent notable differences between the treatments at *p* < 0.05. CK, *Brassica juncea* alone; B, *Brassica juncea* + biochar; T1, *Brassica juncea* + 4.5 g m^−2^ TH; T2, *Brassica juncea* + 9 g m^−2^ TH; T1B, *Brassica juncea* + biochar + 4.5 g m^−2^ TH; T2B, *Brassica juncea* + biochar + 9 g m^−2^ TH.

**Figure 2 plants-12-02939-f002:**
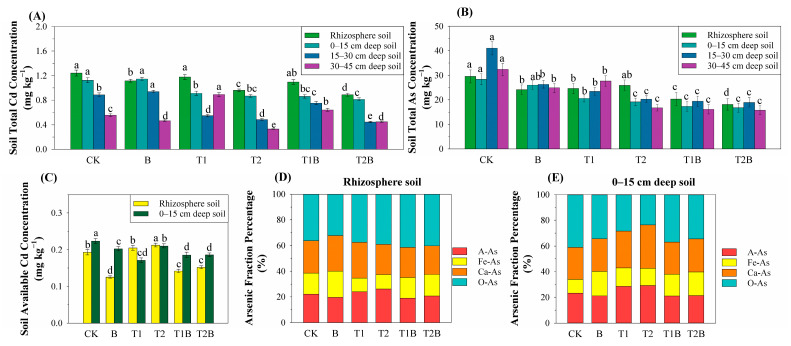
Cd concentration (**A**) and As concentration (**B**) in different soil layers. (**C**) Availability of Cd in the rhizosphere and 15 cm deep soil layer. Morphological classification of soil As in the rhizosphere soil (**D**) and the 15 cm depth soil layer (**E**). Different letters represent notable differences between the treatments at *p* < 0.05. CK, *Brassica juncea* alone; B, *Brassica juncea* + biochar; T1, *Brassica juncea* + 4.5 g m^−2^ TH; T2, *Brassica juncea* + 9 g m^−2^ TH; T1B, *Brassica juncea* + biochar + 4.5 g m^−2^ TH; T2B, *Brassica juncea* + biochar + 9 g m^−2^ TH.

**Figure 3 plants-12-02939-f003:**
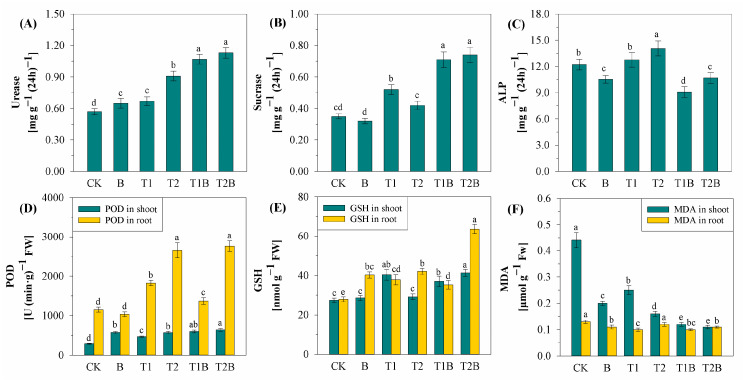
Changes in the activities of enzymes in the rhizosphere soil: urease (**A**), sucrase (**B**), ALP (**C**). Changes in antioxidant activities of plant shoot and root under different treatments: POD (**D**), GSH (**E**), MDA (**F**). Different letters represent notable differences between the treatments at *p* < 0.05. CK, *Brassica juncea* alone; B, *Brassica juncea* + biochar; T1, *Brassica juncea* + 4.5 g m^−2^ TH; T2, *Brassica juncea* + 9 g m^−2^ TH; T1B, *Brassica juncea* + biochar + 4.5 g m^−2^ TH; T2B, *Brassica juncea* + biochar + 9 g m^−2^ TH.

**Figure 4 plants-12-02939-f004:**
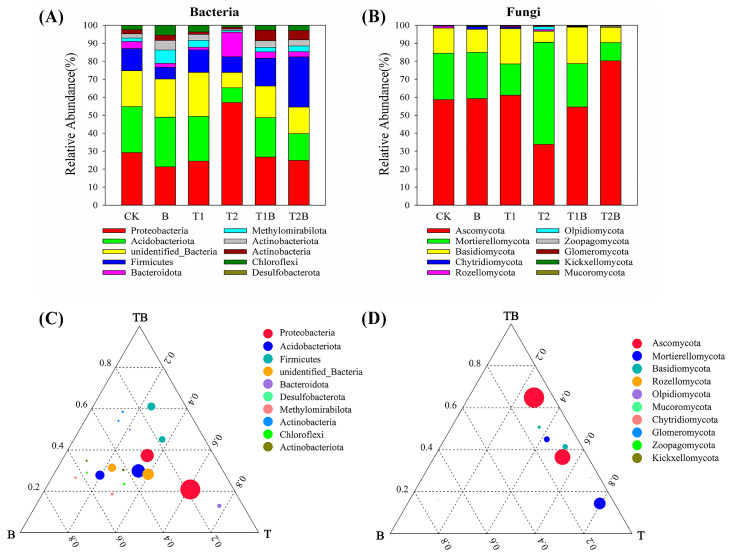
Relative percentage of the top 10 predominant microorganisms of bacteria (**A**) and fungi (**B**) at phylum level (%). Ternary plots of B (B), T (T1, T2), TB (T1B, T2B) samples of the top 10 species in terms of mean abundance at the phylum taxonomic level were selected for analysis: bacteria (**C**) and fungi (**D**). The relative abundance (weighted average) of an OTU is shown by the size of its circle. The specified compartments contribute to the total relative proportion, and this determines the circle’s position. CK, *Brassica juncea* alone; B, *Brassica juncea* + biochar; T1, *Brassica juncea* + 4.5 g m^−2^ TH; T2, *Brassica juncea* + 9 g m^−2^ TH; T1B, *Brassica juncea* + biochar + 4.5 g m^−2^ TH; T2B, *Brassica juncea* + biochar + 9 g m^−2^ TH.

**Figure 5 plants-12-02939-f005:**
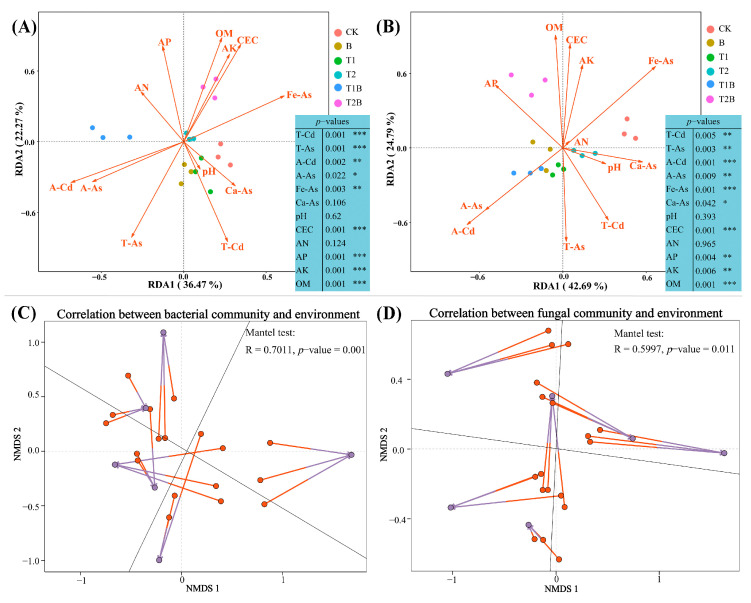
Redundancy analysis of physicochemical properties in rhizosphere soil for bacteria (**A**) and fungi (**B**). Procrustes analysis was performed to investigate the association between environmental factors and all bacterial (**C**) and fungal (**D**) communities, following the NMDS outcomes (using the Bray–Curtis approach). The connection between the composition of the fungal community and environmental factors was demonstrated by the direction and size of the arrows. The ‘*p*-values’ reveal the consequences of significance tests for disparities among environmental factors and soil microbial communities. *p* ≤ 0.001 is represented by three symbols (***), 0.001 < *p* ≤ 0.01 is demonstrated by two asterisks (**), and 0.01 < *p* ≤ 0.05 is demonstrated by a single asterisk (*).

**Figure 6 plants-12-02939-f006:**
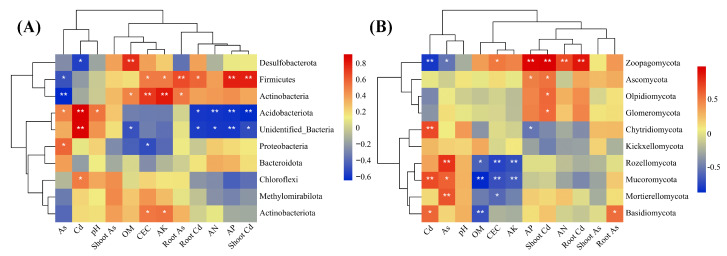
The top 10 most prevalent bacterial (**A**) and fungal (**B**) groups were associated with soil physical and chemical characteristics and plant PTE buildup by the method of Spearman correlation heat map with UPGMA clustering. Blue and red illustrate negative and positive associations, respectively. *p* ≤ 0.01 is demonstrated by two asterisks (**), and 0.01 < *p* ≤ 0.05 is demonstrated by a single asterisk (*).

**Figure 7 plants-12-02939-f007:**
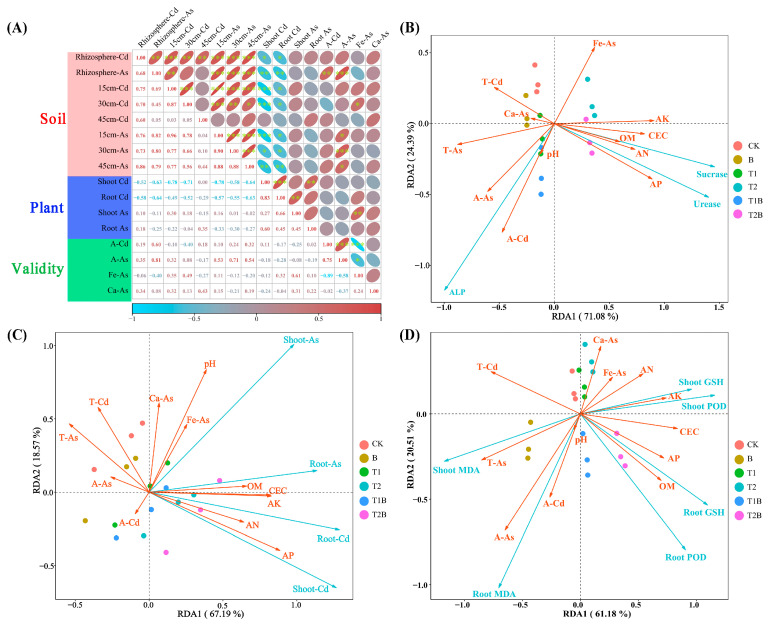
(**A**) Pearson significance analysis of the PTEs in different soil horizons, plant, and the validity of PTEs in rhizosphere soil. (**B**) RDA of soil enzyme operation and soil physical and chemical attributes. (**C**) RDA of plant PTEs accumulation and soil physical and chemical attributes. (**D**) RDA of plant enzyme activity and soil physicochemical properties. *p* ≤ 0.001 is represented by three symbols (***), 0.001 < *p* ≤ 0.01 is demonstrated by two asterisks (**), and 0.01 < *p* ≤ 0.05 is demonstrated by a single asterisk (*).

## Data Availability

The data presented in this study are available on request from the corresponding author.

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
