# Peer review of "Combination of Biochar and Trichoderma harzianum Can Improve the Phytoremediation Efficiency of Brassica juncea and the Rhizosphere Micro-Ecology in Cadmium and Arsenic Contaminated Soil"

_plants, 2023, doi:10.3390/plants12162939_

Round 1
Reviewer 1 Report
The paper is complete and studies very varied aspects. It seems interesting to me because it deals with the subject of phytoremediation with Brassica juncea considering several combined methods such as biochar and the use of Trichoderma harzianum, for which they study physicochemical aspects, enzymatic activities and microbiology, mainly of the rhizosphere.
I am going to make some considerations or comments that I believe can improve the paper:
In the title of the paper and throughout the manuscript heavy metals (HMs) are mentioned, but really only As and Cd are studied, I think it is a bit misleading and that from the beginning the title should reflect that only these two elements are studied and not heavy metals in general.
The problem or the most important shortcoming that I see in the work is the scarcity of samples, a population of 3 samples, n=3, in each treatment seems to me to be too small to be able to reach representative conclusions. I believe that a larger number of samples would be desirable for the statistics to be significant and to reach good extrapolable conclusions.
Regarding the enzymatic activities, why was not beta-glucosidase (from the C cycle) or dehydrogenase considered more generally?
It does not seem to have started from a contaminated soil to decontaminate it because an initial soil content of 1.54 ppm of Cd does not seem to me to be much contamination or 24.42 ppm of As, in a phytoremediation experiment, should we not start from a very contaminated soil?
The concrete mechanisms that are raised in the objectives I believe that they do not end up being specified.
As for more format aspects:
Thichoderma harzianum and Brassica juncea should always be in italics and sometimes they are not in italics, for example in the conclusions or in the supplementary material.
I particularly like the material and methods section after the introduction to see how the work has been carried out and then read the results obtained and not at the end. But in Plants it seems that they are usually put at the end.
To see the statistical differences between groups, sometimes lower case letters are used (a, b, c, d...) and in other figures they are capitalized (A, B, C....). I think it is better to unify and put them all the same, in lower case.
Although I think the figures are very accurate and beautiful. Some figures, I believe that they have the legend too small and it is illegible, for example in the figure S2, the figure 7, the figure 5, etc.
Regarding the bibliography, perhaps it is too extensive 99 references, are they all necessary? The name of the journal should be abbreviated according to the rules of the journal and they are not abbreviated, they should be abbreviated.
Reviewer 2 Report
The manuscript investigated the effects of combined biochar and Trichoderma harzianum on improving the heavy metal phytoremediation efficiency of Brassica juncea and the response of the rhizosphere soil microecology. The experiment was well conducted and the manuscript was well prepared. The results are important in the context of phytoremediating heavy metal contaminated soils. However, several issues need to be addressed before publication.
1. It seems that the soil was severely contaminated by Cd and Hg, and soil As content was not high. Why the authors only focused on Cd and As?
2. More information about the biochar should be provided.
3. Why Trichoderma harzianum promoted the translocation factor of Brassica juncea should be more thoroughly discussed.
Reviewer 3 Report
The manuscript entitled: Effects of combined biochar and Trichoderma harzianum on improving the heavy metal phytoremediation efficiency of Brassica juncea and the response of the rhizosphere soil microecology deals with an environment-friendly method for soil polluted with toxic elements remediation. The article contains interesting data and is well written, but some improvements are still necessary, according to the comments below.
The abstract section should shorten since exceed 200 words, recommended in journal template. The main findings should be presented more concise.
Arsenic is not a heavy metal, strictly considering heavy metals definition. Please avoid use this term and replace by potentially toxic elements (PTEs), including in the Title. The most appropriate term to replace HMs throughout the text is PTEs.
Materials and Methods
Please provide the type of graphite digestion instrument used for soil and plants digestion.
Please provide more details about elements determination by atomic absorption spectrophotometry. It was GFAAS, or FAAS?. Please see for reference:
https://doi.org/10.3390/molecules25112591
Round 2
Reviewer 1 Report
The authors have responded to each of the comments and suggestions I made to them. I believe that the paper has improved considerably and that it can be published in Plants.
Reviewer 2 Report
The authors properly revised the manuscript based on the comments from the reviewers. It can be accepted for publication now.